# Microwave-Driven Electrodeless Ultraviolet Lamp Based on Coaxial Slot Radiator

**Yuqing Huang, Nanya Zhong, Huacheng Zhu *** and **Kama Huang**

College of Electronic and Information Engineering, Sichuan University, Chengdu 610065, China;
huangyuqing@stu.scu.edu.cn (Y.H.); zhongny29@163.com (N.Z.); kmhuang2020@163.com (K.H.)
* Correspondence: hczhu@scu.edu.cn; Tel.: +86-028-5847-0659

**Abstract:** Microwave-driven electrodeless ultraviolet (UV) lamps have the advantages of high efficiency and high power. However, the conventional microwave system is slightly oversized, which restricts the use of the lamp in a narrow space. A miniaturized microwave-driven electrodeless UV lamp based on a coaxial slot antenna was developed in this study. First, the structure of slots was optimized using a finite-difference time-domain algorithm such that high efficiency of radiated energy could be achieved. Second, a complex model based on the Drude model and the electromagnetic theory was established to simulate the interaction between the microwave and UV lamps. The efficiency and uniformity of the UV lamps were analyzed. Finally, an experimental system was built, and the computed results agreed well with the simulation results. The efficiency of the miniaturized microwave-driven electrodeless UV lamp reached 91.8%.

**Keywords:** electrodeless UV lamp; microwave plasma; Drude model; miniaturization

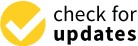



## 1. Introduction

Sterilization is an essential procedure in medical treatment, environmental protection, food safety, and reuse of water. At present, ultraviolet lamps are widely used in various industries as a general sterilization method, but the UV lamp has the disadvantages of insufficient power and incomplete sterilization [1]. The microwave-driven electrodeless ultraviolet lamp provides a new solution for sterilization. In 2016, Zhang et al. proposed a microwave-induced electrodeless ultraviolet (MW-EUV) lamp and demonstrated that MW-EUV irradiation was a rapid and efficient method without photoreactivation. The treated water met the hygiene standards for the reuse of recycled water [2]. Furthermore, electrodeless UV lamps can be utilized in air disinfection as purifiers [3]. Subsequently, UV lamps have a wide range of uses in chemical fields [4–9].

The use of microwave-driven electrodeless UV lamps, which have significant advantages in sewage treatment, has been proposed. These lamps have a long life, high power, and high efficiency. Additionally, there are no complications in lamp shape and no variations in light intensity (the light output is stable), indicating that rapid and complete sterilization can be achieved [10–13]. The initial microwave-driven electrodeless UV lamp lies inside the waveguide. This structure can be regarded as a microwave radiator that excites plasma to emit UV light. Due to the work of the waveguide requiring an anhydrous environment, the waveguide is surrounded by a quartz column to make sure that the entire structure represents a sealed environment for underwater sterilization [14,15]. Horikoshi et al. proposed a structure in which microwave radiation is used as a heat source, and UV and microwave radiation are available simultaneously [16].

UV lamps can be widely used in various fields. For instance, they can be used for chemical synthesis, warnings of approaching missiles, and sterilization. Because of the characteristics of microwave-driven electrodeless UV lamps, several related studies have been conducted. Furthermore, Remya and Swain investigated the degradation of soft drink industrial wastewater using microwave electrodeless UV lamps [17]. In addition,

Horikoshi et al. proposed a new type of microwave electrodeless UV lamp structure and conducted relevant research on the chemical oxidation and photolysis of contaminated water [16]. Shi and Chen studied the performance of microwave electrodeless lamps at high frequency and compared electrodeless microwave lamps with conventional Hg lamps [18,19]. Horikoshi et al. also considered the efficiency of photo-assisted degradation of organic substrates using a microwave discharge electrodeless lamp in which $TiO_2$ is present, which can assist in the photodegradation of a large class of organic pollutants. The degradation reached 100% within 10 min [20–22]. Oberreuther et al. examined a dry plasma-reforming process of carbon dioxide with methane, which runs in microwave plasma under atmospheric conditions. The efficiency reached 90–100% [23]. Conventional microwave electrodeless UV lamps use rectangular waveguides to couple energy, which are difficult to miniaturize due to size constraints. The structure of the rectangular waveguide leads to uneven distribution of the energy and different brightness of the UV lamp. Zhong et.al. proposed a microwave-driven UV lamp with a rectangular structure to solve the uneven distribution of light intensity [24]. However, most of these devices are significantly large and thus cannot be used in some narrow spaces. In areas such as places with pipelines, most of the equipment cannot be used, and devices that have been miniaturized are generally limited by their low power and poor uniformity.

Some research has been conducted to miniaturize radiators. Zhuge et al. considered a miniaturized dielectric resonator for microwave plasma lamps. This structure reduces the weight of the dielectric cavity and thus the weight of the plasma lamp [25]. Wu et al. proposed a miniaturized coaxial slot radiator structure and studied the effect of microwave and UV on epoxy curing. The results showed that the use of the MW combined with UV method can make the epoxy resin cure more thoroughly and have better adhesion. However, the microwave input power is only 100 W, and there is strong microwave radiation [26]. Kando et al. investigated a new type of high-pressure microwave discharge for delivering microwave power to the center of compact high-pressure discharge lamps [27].

The waveguide slot radiator is realized by adding a narrow slot on the sidewall of the waveguide. It has the characteristics of stable performance, solid structure, high power bearing capacity, and low loss [28]. Zhao et al. considered an eight-element waveguide slot array based on rectangular waveguide. The measured bandwidth for VSWR $\leq$ 1.5 is 19.6% [29]. To improve the bandwidth of slot waveguide antenna, Wang et al. investigated a longitudinally slotted ridge waveguide and the measured bandwidth of VSWR $\leq$ 1.43 has reached 14.9% [30]. Shin et al. proposed the cylindrical slot waveguide radiator and studied its radiation properties. The measured bandwidth of S11 $\leq$ $-10$ dB is 15.2% [31]. Ho et al. also investigated an antenna based on a cylindrical waveguide and a helical slot [32].

In this study, a miniaturized microwave-driven electrodeless UV lamp based on a slot was developed. It solves the problem of device size and achieves the miniaturization of the device without being limited by low power [33]. Using a coaxial slot structure, the scope of the device was narrowed without reducing the power of the UV lamp. First, the finite-difference time-domain (FDTD) algorithm was utilized to optimize slots to achieve high efficiency of radiated energy. Section 2 presents a model established by combining the Drude plasma model [34] and electromagnetic theory to compute the interaction between the microwave and UV lamps. In addition, an experimental system was built to verify the performance of the miniaturized device. Finally, the sensitivity of the structure is discussed.

## 2. Methodology

Aiming at the problems of the traditional electrodeless UV lamp proposed above, we constructed and optimized the model of the miniaturized electrodeless UV lamp by using the finite-difference time-domain algorithm.

### 2.1. Geometry

As shown in Figure 1, the system used in this study is based on a coaxial slot waveguide. We calculated the model with three, five, and eight slots, respectively. The electric field distribution of the structure is the most uniform when the number of slots is eight, as shown in Figure 2. There are eight slots around the coaxial waveguide, and a hollow ring quartz tube is sleeved outside the coaxial radiator. The quartz tube is covered with metal mesh, not depicted in Figure 1, to prevent electromagnetic waves from leaking, and the tube is mainly filled with argon and liquid Hg. Electromagnetic waves are coupled in open slots in the coaxial waveguide to excite the plasma lamp. A 2D top view, side view, and cross section of the structure are illustrated by Figure 3.

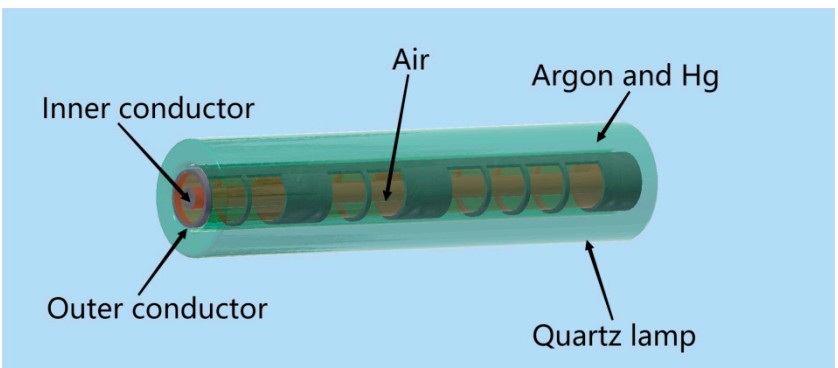

**Figure 1.** Diagram of slotted coaxial waveguide.

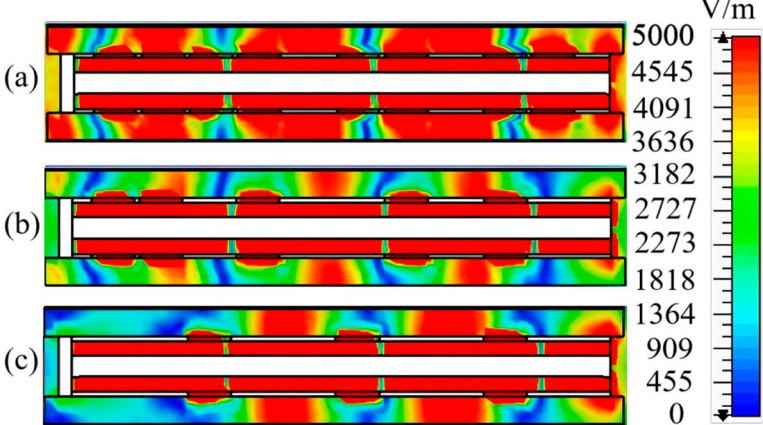

**Figure 2.** Electric field distributions with different number of slots: (**a**) 8, (**b**) 5, and (**c**) 3.

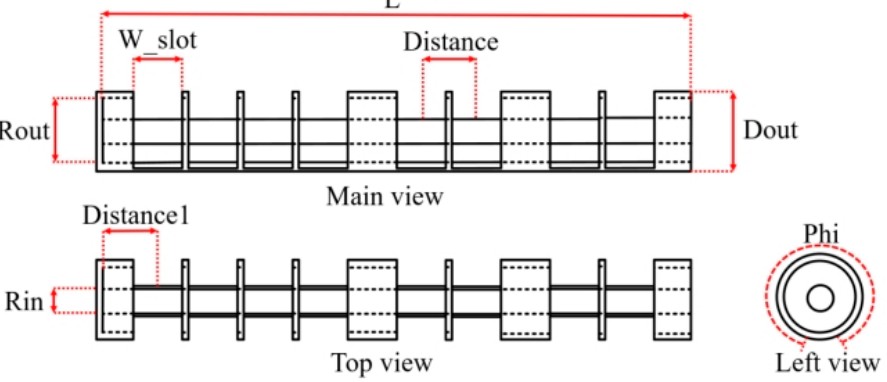

**Figure 3.** 2D top view, side view, and cross section of the structure.

*2.2. Plasma Parameters*

At a frequency of 2450 MHz, the change in the electron mobility so small that can be ignored. The plasma parameters can be considered as constants.

In this study, the relative permittivity of a dispersive medium was calculated from the Drude model. In this model, the collision frequency $v_m$, electron plasma frequency $\omega_{pe}$, and breakdown field $E_c$ are three critical parameters.

Firstly, the electron plasma frequency $\omega_{pe}$ can be determined using Equation (1):

$$\omega_{pe} = \sqrt{\frac{n_e e^2}{\varepsilon_0 m_e}}, \tag{1}$$

where $\varepsilon_0$ is the vacuum permittivity, $e$ is the element charge, $m_e$ is the mass of the element charge, and $n_e$ is electron density. Second, the collision frequency $v_m$ is expressed by Equation (2):

$$v_m = \frac{\sqrt{8K_B T_e / \pi m_e}}{\lambda_e}, \tag{2}$$

where $K_B$ is the Boltzmann constant, $T_e$ is the electron temperature, and $\lambda_e$ shows the mean free path between the neutral atoms and electrons. One can ignore the collision loss for argon gas if the microwave frequency is significantly larger than the collision frequency.

$$v_m \approx 1.52 \times 10^7 p \sqrt{T_e}, \tag{3}$$

$$\varepsilon_p = \varepsilon_0 \left( 1 - \frac{\omega_{pe}^2}{\omega(\omega - jv_m)} \right) = E'_{ps} - E''_{ps}, \tag{4}$$

where $p$ is the pressure, the permittivity of plasma is expressed in terms of $\omega_{pe}$ and $v_m$. $E'_{ps}$ and $E''_{ps}$ are the real and imaginary parts of the permittivity, respectively. As shown in Figure 4, the dispersion curve was calculated from $\omega_{pe}$ and $v_m$.

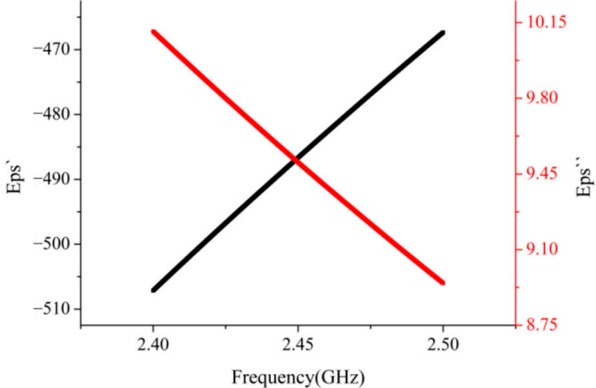

**Figure 4.** Dispersion curve.

The breakdown field strength $E_c$ can be expressed by Equation (5):

$$E_c = \frac{K_B T \omega}{pS \wedge \sqrt{\frac{m_e v_i}{3e}}}, \tag{5}$$

$$\wedge = \frac{1}{\sqrt{\left(\frac{\pi}{l}\right)^2 + \left(\frac{2.405}{R}\right)^2}}, \tag{6}$$

where $l$ is the length of tube, $R$ is inner diameter, $S$ is the elastic collision cross-section, $T$ is the gas temperature, and $v_i$ is the first ionization energy of neutral particle [24].

Only when the incident power of the microwave is sufficiently high, causing its electric field strength to exceed the breakdown field strength of argon, can the UV lamp be excited. The change in microwave incident power has a minimal effect on the electric field distribution, and it only changes the amplitude of the electric field. The breakdown field strength of argon under this structure, determined from previous calculations, is 3965 V/m.

### 2.3. Input Parameters

The parameters used in the simulations for the microwave and plasma are listed in Table 1. Both the inner and outer conductors of the coaxial waveguide are considered perfect electrical conductors (PEC).

**Table 1.** Input parameters.

| Parameter | Value |
|---|---|
| Input power | 500 W |
| Microwave frequency | 2450 MHz |
| Plasma frequency | $3.4 \times 10^{11}$ rad/s |
| Collision frequency | $3 \times 10^8$ Hz |
| Plasma breakdown field strength | $3.965 \times 10^3$ V/m |
| Plasma frequency maintained | $1.7 \times 10^{11}$ rad/s |

### 2.4. Boundary Conditions

In this study, the entire structure included a coaxial radiator matched to a coaxial cable with a characteristic impedance of 50 $\Omega$. In the simulations, because the radiator was made of metal, it was assumed to be a perfect electrical conductor that included a metal inner conductor, metal outer conductor, metal short-circuit surface of the terminal, and metal mesh covering the hollow ring quartz tube. A metal mesh was designed to prevent electromagnetic waves from leaking. Working at 2.45 GHz, the microwaves corresponded to a wavelength of approximately 122.44 mm. If the value is significantly higher than the gap of the metal mesh, electromagnetic waves cannot escape.

The boundary conditions for a perfect electrical conductor can be expressed by Equation (7):

$$\vec{e_t} \times \vec{E} = 0,$$
$$\vec{e_n} \times \vec{H} = 0, \tag{7}$$

where $\vec{e_t}$ is the tangential component, $\vec{e_n}$ is the normal component.

The material of the electrodeless UV tube between the metal mesh and coaxial radiator was a quartz wall. Quartz had a relative dielectric constant of 4.2 and did not have an imaginary part, suggesting that the effect of the quartz wall on electromagnetic waves was negligible.

## 3. Results and Discussion

### 3.1. Dimension Optimization

The dimensions of this new structure were determined by three factors: the field uniformity, reflection coefficient (S11), and electric field strength. Based on S11, one can observe the incident power and reflected power of this new microwave electrodeless UV device. In general, the magnitude of S11 is preferably lower than −10 dB. Then, the strength of the electric field around the UV lamp was the next point to consider. Finally, it is necessary to consider whether the coupling energy of eight slots is uniform, which is critical to ensure that the radiation of the hollow ring quartz tube is uniform in all directions.

The lengths of the coaxial radiator and ring tube were both 200 mm. The outer and inner diameters of the tube were 45 and 25 mm, respectively. The designed coaxial radiator

was inserted into the tube. The device uniformly coupled microwave energy from the radiator into the tube by coupling through gaps and radiating the UV light. After ensuring the length, one must ensure that the outer and inner diameters of the conductor are correct. These parameters are given by Equation (8).

$$Z_0 = \frac{60}{\sqrt{\varepsilon_r}} \ln \frac{D}{d}, \tag{8}$$

Here, $\varepsilon_r$ is the relative permittivity between the inner conductor $d$ and outer conductor $D$. Air was selected as the dielectric, with a relative permittivity of 1. Moreover, the final outer and inner diameters of the conductor were 19 and 8 mm, respectively, to match the characteristic impedance of the coaxial radiator with the 50 Ω coaxial line widely used in the industry and laboratories. As expressed by Equation (8), the characteristic impedance of the coaxial radiator was approximately 50 Ω; unless the characteristic impedance matches well with the coaxial feeder impedance, there are reflections due to impedance mismatching.

After the above parameters were determined, the FDTD method was applied to emulate the entire model structure. In this simulation, the values of the two parameters, W_slot and phi, were analyzed. These parameters were calculated and optimized separately. The two steps are considered below.

First, eight gaps were slotted in the radiator with a total length of 200 mm. These gaps had the same width and central angle. It was presumed that the distance between adjacent slotted centers was 20 mm, and the distance from the short-circuit surface to the first slotted center was 15 mm to determine whether the values for S11 were lower than −10 dB and to determine the optimal width and central angle. After the optimal width and central angle of the slots were obtained, the distance between adjacent slots was verified, and three parameters—the value of S11, uniformity of the electric field distribution, and magnitude of the electric field—were observed to determine the optimal dimensions. The above calculations were based on the Drude model and FDTD algorithm.

The effect of W_slot, the width of all slots, on S11 for an operating frequency of 2.45 GHz is shown in Figure 5. S11 varied with the change in slot width. Under different width parameters, the microwave attained resonance at the operating frequency, and S11 increased with increasing W_slot. When the value of W_slot was 17 mm, S11 exceeded −10 dB, indicating that only the width between 9 and 16 mm could satisfy the requirement of S11. The slot can couple microwave energy because the slot cuts off the current such that microwave energy is coupled in the longitudinal direction of the slot. Therefore, the more the power line is cut, the stronger the coupled energy. The position of the slots also influences the coupling energy. Based on the principles above, the optimal value of W_slot was 16 mm, and W_slot was set as a constant in the subsequent calculations. At this time, S11 was determined to be −10.25 dB, and the absorption efficiency reached 90%.

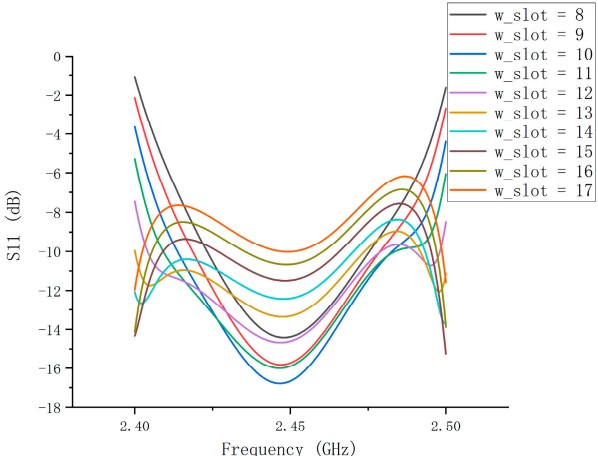

**Figure 5.** Comparison of S11 among different W_slot values.

As shown in Figure 6, the effect of the angle on S11 was analyzed after the widths of the slots were determined. There were ten datasets to satisfy the requirements of S11 at 2.45 GHz. The energies coupled in different central angles were compared and it was determined that the best value for phi was 310°. Thus, phi was set as 310° in the subsequent calculations.

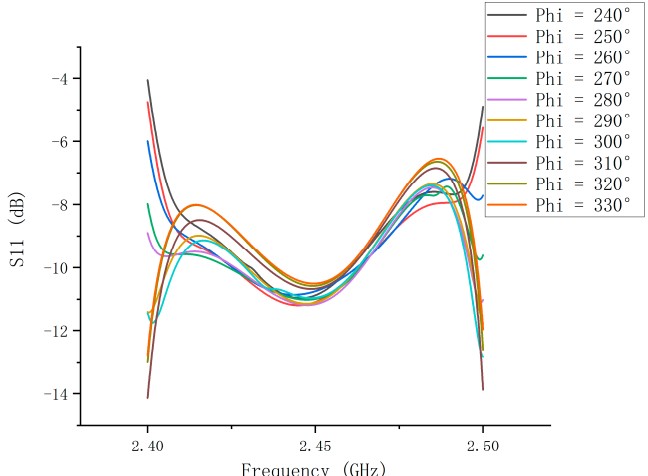

**Figure 6.** Comparison of different phi values of S11.

After the dimensions of the slots were determined, the FDTD algorithm and Drude model were used to calculate the optimal distance between the eight slots. For the coaxial slot microwave electrodeless UV radiator used in this study, the microwave feed port was at the end of the radiator, and the terminal short-circuit surface was at the other end. Microwaves were fed from the microwave feed port. Because the slots had the same size, the closer the slot is to the feed port, the greater the amount of energy coupled, resulting in an uneven electric field distribution in the axial direction of the entire radiator. Therefore, the slots were divided into three groups with four, two, and two slots. The group with four slots was set at the end from the feed port, and the distance between adjacent slotted centers was determined to be 18 mm. The distance for each group was adjusted to obtain the final optimization results. As shown in Figure 7, S11 was approximately −12 dB at a frequency of 2.45 GHz, and the power absorption rate reached 93.7%. At this point, all initial dimension parameters had been obtained (Table 2).

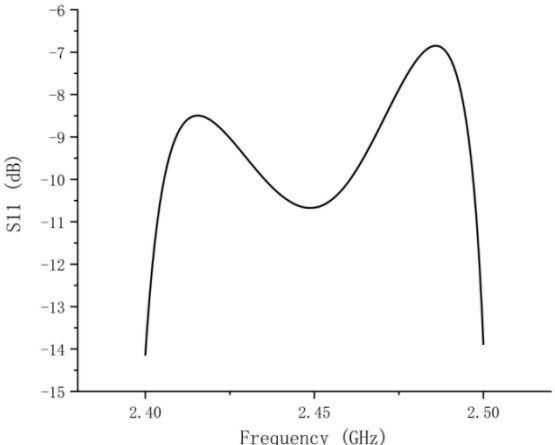

**Figure 7.** S11 of optimized model.

**Table 2.** Optimized dimensions of experimental equipment.

| Parameter | Physical Meaning | Value |
|---|---|---|
| W_slot | Width of all slots | 16 mm |
| Phi | Central angle of all metal slots | 310° |
| $R_{in}$ | Outer diameter of inner conductor of coaxial radiator | 8 mm |
| $R_{out}$ | Inner diameter of outer conductor of coaxial radiator | 19 mm |
| l | Total length of coaxial radiator | 200 mm |
| Distance | Distance between adjacent slotted metal centers | 18 mm |
| Distance 1 | Distance from first slotted center to short-circuit surface | 15 mm |
| $D_{out}$ | Outer diameter of outer conductor of coaxial radiator | 25 mm |

After sizing the model, we calculate the electric field strength and electric field distribution using the FDTD algorithm. Figure 8a shows the three-dimensional electric field distribution, and the two-dimensional electric field distribution cut along the coaxial radiator is shown in Figure 8b, where the maximum value is set to 5000 V/m. The covariance (COV) of the electric field strength is introduced to describe the uniformity of the electric field, and the COV can be expressed by Equation (9):

$$COV = \sqrt{\frac{\sum_n (E_i - \overline{E})^2}{n}}/\overline{E},\tag{9}$$

where $n$ is the total number of points in the selected region, $E_i$ is the electric field value at the $i$-th point of the selected region, and $\overline{E}$ is the average electric field strength of the selected region [25]. The smaller the covariance value, the more uniform the electric field distribution.

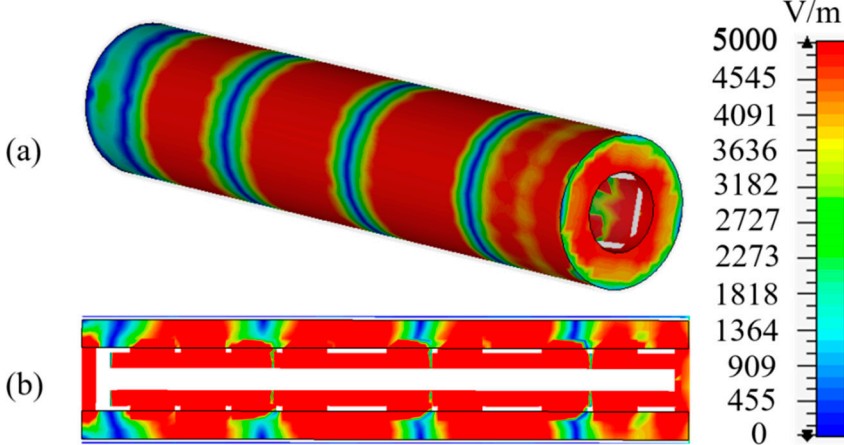

**Figure 8.** Electric field distribution of optimized structure: (**a**) three-dimensional electric field distribution, (**b**) two-dimensional electric field distribution cut along the coaxial radiator.

Figure 8 shows that the electric field distribution is uniform in the axial direction and all angles in the radial direction. The COV is 0.38 and the electric field can be considered relatively uniform.

### 3.2. Sensitivity Analysis

The frequency and collision frequency are two important parameters for plasma. First, it was assumed that the microwave power was 500 W, adopted as a constant, and the collision frequency of the plasma was $3 \times 10^8$ Hz. The plasma frequencies were set as $3.3 \times 10^{11}$, $3.4 \times 10^{11}$, and $3.5 \times 10^{11}$ rad/s. The electric field distributions at different frequencies are shown in Figure 9. A slight change was observed with the variation in the plasma frequencies, and the maximum and average values of the electric field are shown

in Table 3. Overall, the electric field distribution at different plasma frequencies (Figure 9) shows a uniform and periodic distribution in the axial direction. In the 360° radial direction, the electric field distribution was also close to uniform. As shown in Figure 10, S11 exhibits almost the same trend with different plasma frequencies. The values of S11 were lower than −10 dB at 2.45 GHz. Consequently, when the plasma frequency changed, the reflected power was somewhat low, and almost no deviation was observed. The above results confirm that this structure is applicable to different lamp types.

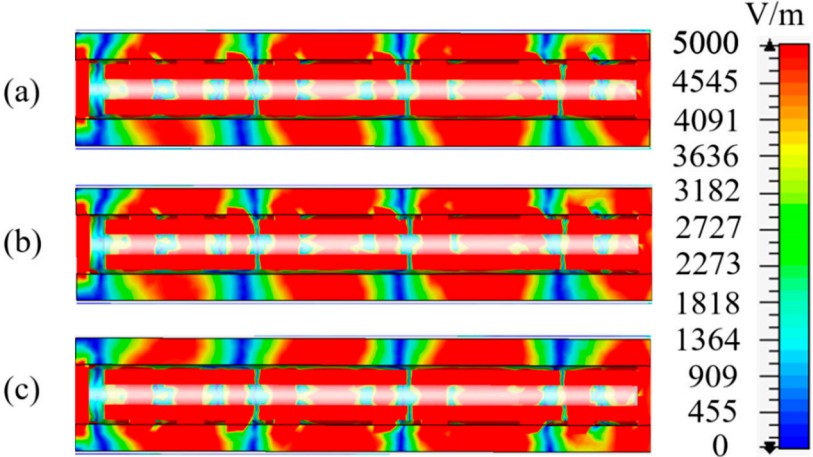

**Figure 9.** Electric field distributions at different plasma frequencies: (**a**) $3.3 \times 10^{11}$ rad/s, (**b**) $3.4 \times 10^{11}$ rad/s, and (**c**) $3.5 \times 10^{11}$ rad/s.

**Table 3.** Maximum and average values of the electric field at different plasma frequencies.

| Plasma Frequencies | Maximum | Average |
| --- | --- | --- |
| $3.3 \times 10^{11}$ rad/s | 121,850 V/m | 77,754 V/m |
| $3.4 \times 10^{11}$ rad/s | 122,489 V/m | 77,980 V/m |
| $3.5 \times 10^{11}$ rad/s | 122,109 V/m | 77,854 V/m |

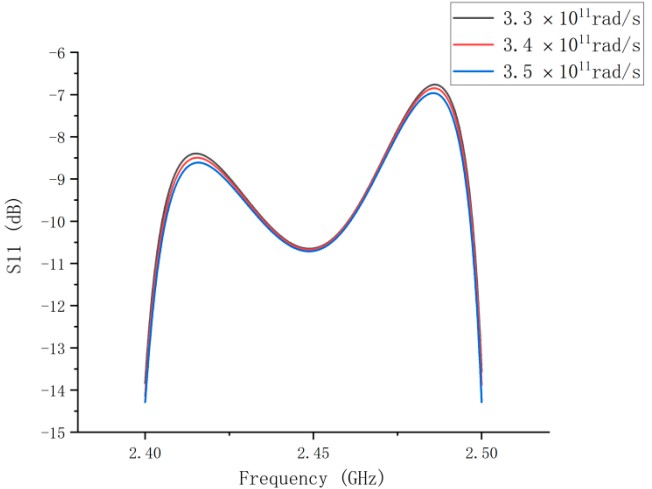

**Figure 10.** Comparison of S11 with different plasma frequencies.

The plasma frequency was assumed to be $3.4 \times 10^{11}$ rad/s, and the collision frequency values of the plasma were set as $2.8 \times 10^8$, $3.0 \times 10^8$, and $3.2 \times 10^8$ Hz. The electric field distributions for different plasma collision frequencies are shown in Figure 11. When the plasma collision frequency changed, mainly because of the variation in the relative

permittivity, the electric field distribution exhibited a slight change, and the maximum and average values of the electric field are shown in Table 4. The electric field is periodically distributed in the axial direction. In the radial direction, the electric field distribution is also close to uniform.

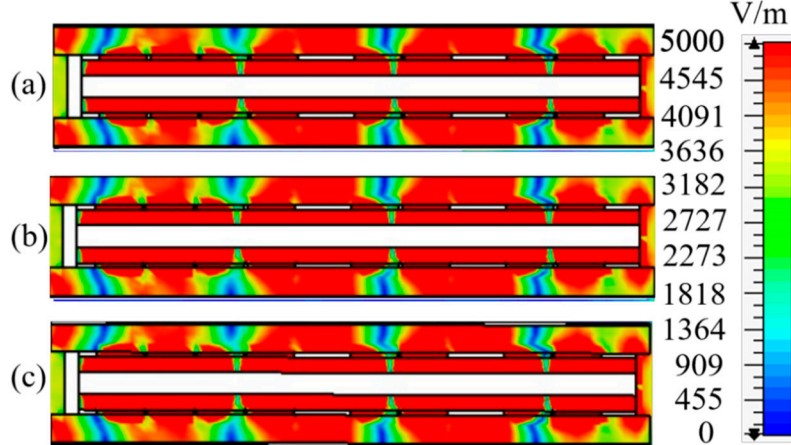

**Figure 11.** Electric field distributions at different plasma collision frequencies: (**a**) $2.8 \times 10^8$ Hz, (**b**) $3.0 \times 10^8$ Hz, and (**c**) $3.2 \times 10^8$ Hz.

**Table 4.** Maximum and average values of the electric field at different plasma collision frequencies.

| Plasma Collision Frequencies | Maximum | Average |
|---|---|---|
| $2.8 \times 10^8$ Hz | 122,050 V/m | 77,702 V/m |
| $3.0 \times 10^8$ Hz | 122,489 V/m | 77,980 V/m |
| $3.2 \times 10^8$ Hz | 122,195 V/m | 77,796 V/m |

S11 exhibits almost the same trend in each case as shown in Figure 12. The values of S11 were all less than $-10$ dB at 2.45 GHz. When the plasma collision frequency changed, the power absorption did not deteriorate. One reason is that the change in plasma collision frequency influences both the imaginary and real parts of the relative dielectric constant, ultimately resulting in no significant change in the loss tangent. Consequently, S11 does not deteriorate. Another reason is that we completed the plasma frequency sensitivity analysis, demonstrating that the designed structure can be applied to various lamps. Figure 11 shows that the designed system can be used on a UV lamp with frequencies of $2.8 \times 10^8$, $3.0 \times 10^8$, and $3.2 \times 10^8$ Hz.

### 3.3. Experimental Verification and UV Light Power Measurement

The overall diagram of the experimental system is shown in Figure 13. The main function of the solid-state source is to generate microwaves with a maximum output power of 1 kW. The circulator is a device that transmits electromagnetic waves in a one-way ring shape. This device helps protect the expensive microwave generator. The EIT UV power puck is a device that can measure UV power. Three test points are selected in the tangential direction of the lamp, the power is measured by changing the distance between the test point and the lamp, and finally the average value is taken and marked in Table 5. Sterilization can be achieved when the UV light energy is greater than 30 mJ/cm². Energy can be calculated by $E(\text{mJ/cm}^2) = P(\text{mW/cm}^2) \times t(s)$, where P is the power, and t is the time. From Table 5, it is found that sterilization can be complete within 10 s at a position within 15 cm of the UV lamp. As the distance becomes longer, the time required for sterilization increases.

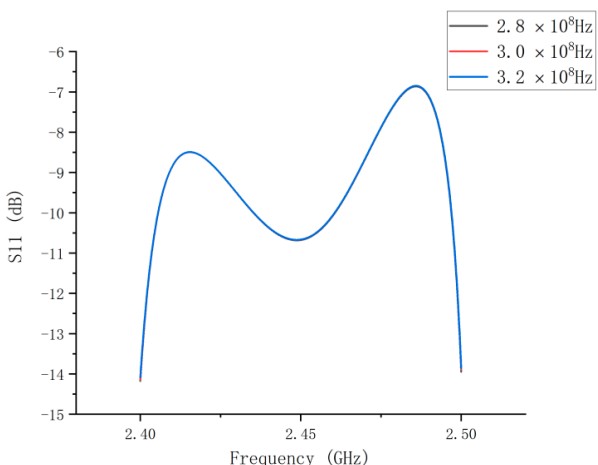

**Figure 12.** Comparison of S11 with different plasma collision frequencies.

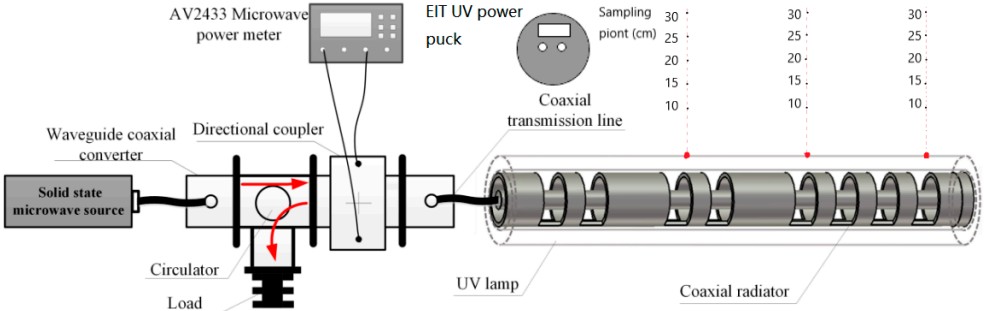

**Figure 13.** Microwave-driven UV lamp block diagram.

**Table 5.** Comparison of UV lamp power with distances.

| Distance | UV Lamp Power |
|:---:|:---:|
| 10 | 6.6236 mW/cm$^2$ |
| 15 | 4.0012 mW/cm$^2$ |
| 20 | 1.7196 mW/cm$^2$ |
| 25 | 1.428 mW/cm$^2$ |
| 30 | 1.2984 mW/cm$^2$ |

The light from the lamp is uniform and stable, as shown in Figure 14. In the hot state, S11 reached −12.2 dB in simulation and −10.9 dB in the experiment. The power efficiency was up to 91.8% in experiment. The comparison results for S11 under hot and cold conditions are shown in Figure 15. The experimental and simulation results were consistent.

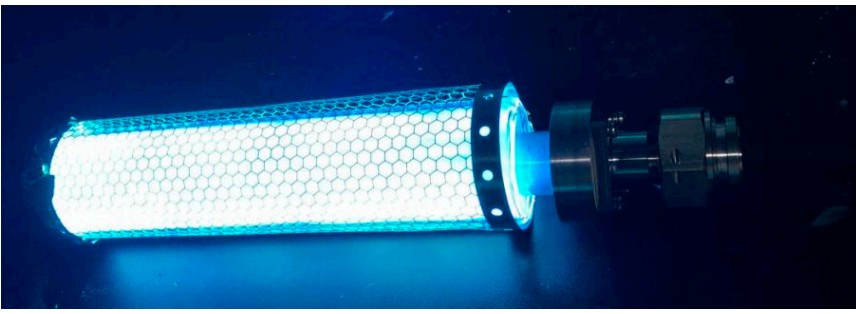

**Figure 14.** Experimental setup.

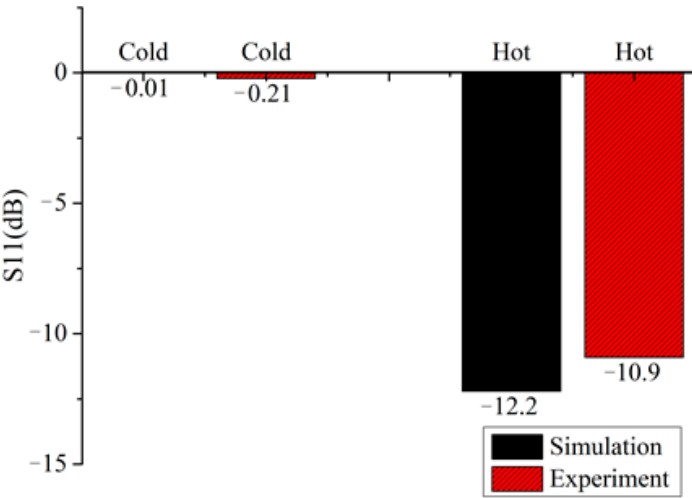

**Figure 15.** Comparing the S11 in different states.

## 4. Conclusions

In this study, we proposed a miniaturized microwave-driven electrodeless UV lamp based on a coaxial slot antenna. The FDTD algorithm was utilized to optimize the entire structure for high efficiency of radiated energy, and the Drude model and electromagnetic theory are used to build complicated models that simulate the interaction between microwaves and UV lamps. For verification, an experimental system was built, and the microwave-driven UV lamp was successfully lit. The electric field distribution was uniform in the axial and radial direction, verifying that the energy of the radiator was uniformly coupled. In the hot state experimental, the power absorption rate could exceed 91%. The miniaturized microwave-driven electrodeless UV lamp can be used in a narrow environment.

**Author Contributions:** Writing—original draft preparation, Y.H.; writing—review and editing, H.Z.; visualization, N.Z.; project administration, H.Z.; funding acquisition, H.Z.; supervision, K.H. All authors have read and agreed to the published version of the manuscript.

**Funding:** This research was funded by key technology research projects in Shunde District, Foshan City, grant number 2130218002514.

**Institutional Review Board Statement:** Not applicable.

**Informed Consent Statement:** Not applicable.

**Data Availability Statement:** Not applicable.

**Conflicts of Interest:** The authors declare no conflict of interest.

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
