# Peer review of "Microwave-Driven Electrodeless Ultraviolet Lamp Based on Coaxial Slot Radiator"

_processes, doi:10.3390/pr10050890_

Round 1

Reviewer 1 Report

Line 60 - Discuss the attempts made to minimize the size and drawbacks observed by other methods and researcher

74- Discuss the advantages of slot structure and also describe various slot strucuture

77- Why 8 slot structure ?  what is the significance of number of slots ?

133- why these three factors are picked ?

192 - Any other criteria to make uniform electric field distribution ?

239 - Any backup for your hypothesis on the changes in power absorption - citation ?

In conclusion, explain in detail regarding your key findings and include the cost efficiency as well

Reviewer 2 Report

I suggest removing or brightening the background of Figure 1.

In line 93, the equation number (3) should be replaced by (4).

The description of Fig. 3 should be improved: the font seems too large.

Comments on the FDTD software used for the simulation would be welcome. The description of the electrical parameters of quartz in the last paragraph of section 2.3 seems a bit too simple: the quartz layer will certainly affect the electric field distribution in the device, since its relative permeability (4.2) is significantly different from that of argon or air (1).

What gas filled the structure? In section 2.2 the authors write about argon, but in section 31 (line 148) they write about air.

I would greatly appreciate a summary of the results of the sensitivity analysis in a table. For example, the comparison shown in Figure 10 seems to show a significant difference between (a) and (b), but in the text the authors use the phrase "small change". Can this "slight difference" be expressed in numbers? The min, max, avg values and possibly the standard deviation...

Reviewer 3 Report

The authors in this work present a miniaturized microwave-driven electrodeless UV lamp based on a coaxial slot antenna for sterilization. The work is interesting, it has however some issues.

  1. Lines 25,40,45, etc.: add “et al.” to the 1st author name.
  2. Across the manuscript, FDTD algorithm is used. Is it authors’ code or some well know software?
  3. Section 2.1 Geometry. The geometry should be presented in detail. Why is this geometry selected? The inner conductor is connected to the right end of the lamp (Fig. 2), the waveguide is filled with Ar, etc.

  1. As an overall comment, the overall geometry and structure seems to be selected and the text slowly reveals information about the why and how each decision was made. This approach is counter-intuitive and keeps the reader with more and more questions. Authors should reform the paper, starting from the Electromagnetics of the problem , then proceed with the general geometry and finally present their optimization process. Last the validation via measurement should be presented.

  1. Given that miniaturization seems to be a core issue for this paper, a comparison of the dimensions for the relevant work as well as this lamp should be highlighted to show the improvement.

  1. Line 80-81 and Figure 2 Caption: rephrase “Two-dimensional views” to something along the lines of “2D top view, side view and cross section of the structure” for clarity.
  2. Figure 1 & 2: they are supposed to depict the 3d and 2d forms of the same geometry, however in Figure 2 both ends seem to have the same width.
  3. Line 78: “The quartz tube is covered with metal mesh” add “not depicted in Figure 1” for clarity.
  4. Line 110: calculation of breakdown field strength under this structure or reference to previous calculations must be provided.
  5. Line 153: “many microwaves are reflected” rephrase to something along the lines of “there are reflections due to impedance mismatching”.
  6. Line 158: there seems to be a word missing after radian or better yet you should not use radian as the word describing the arc removed from the cylindrical waveguide to form the slot. It is a unit after all.
  7. Table 2: replace word “Sign” with “Parameter”
  8. According to Horikoshi [21] :

“Microwave discharge electrodeless lamps (MDELs) are useful light sources for the decontamination of polluted water because UV and microwave radiation can be simultaneously used to induce degradative processes to destroy organic compounds and microorganisms.”

With this form of lamp, the microwave portion is omitted. Is the UV light enough for decontamination? Other works (eg. [8]) used solvents, and high temperatures to help with the process. A discussion is necessary, given the scope of the lamp.

What is the (UV) power of the lamp? Does the metal mesh effect this? The efficiency term is simply the S11 (microwave power delivered to the load). How do this work overcome the problems stated in the introduction

“UV lamp has the disadvantages of insufficient power and incomplete sterilization [1]”. A discussion is necessary, given the scope of the lamp.

  1. Line 280: remove “Please add:”

Reviewer 4 Report

See attached pdf document with comments

Round 2

Reviewer 3 Report

I would like to thank the authors for their response to my comments and the revised version of their manuscript.

They have addressed the majority of my comments.

Regarding the authors' response to the reviewer’s comment 13 :

“Is the UV light enough for decontamination?“

The response to this answer (achieved power levels with decontamination levels and times) along with the measurement process, should be in the paper.

Other comments/corrections

Line 65. “..has been down…”. The expression is idiomatic and should be avoided. Maybe the authors mean “done” instead of “down”?

Line 80 VSWR is a ratio, remove dB

Line 128 change the wording “expressed by” to “depends on” or “expressed in terms of”.

Line 193 The replacement of radian with angular is not correct. Angular is an adjective, not a noun. Replace angular with central angle.

Lines 196 (2 instances), 216 & table 2: replace radian with central angle.

Line 320 I don’t understand the expression “routine” for the “The electric field distribution”. I believe it needs a clearer phrasing.
